# UPLC-Q-TOF/MS Based Plasma Metabolomics for Identification of Paeonol’s Metabolic Target in Endometriosis

**DOI:** 10.3390/molecules28020653

**Published:** 2023-01-09

**Authors:** Jing Liu, Dongxia Yang, Chengyu Piao, Xu Wang, Xiaolan Sun, Yongyan Li, Shuxiang Zhang, Xiuhong Wu

**Affiliations:** 1College of Pharmacy, Heilongjiang University of Chinese Medicine, Harbin 150040, China; 2Department of Gynecology Medicine, Second Affiliated Hospital of Heilongjiang University of Chinese Medicine, Harbin 150001, China; 3Good Laboratory Practice of Drug, Heilongjiang University of Chinese Medicine, Harbin 150040, China

**Keywords:** endometriosis, paeonol, metabolomics, biomarkers, network pharmacology, genes

## Abstract

Endometriosis is a common gynecological illness in women of reproductive age that significantly decreases life quality and fertility. Paeonol has been shown to play an important part in endometriosis treatments. Understanding the mechanism is critical for treating endometriosis. In this study, autologous transplantation combined with a 28 day ice water bath was used to create a rat model of endometriosis with cold clotting and blood stagnation. The levels of estradiol and progesterone in plasma were detected by ELISA, and the pathological changes of ectopic endometrial tissue were examined by H&E staining, which proved the efficacy of paeonol. For metabolomic analysis of plasma samples, UPLC-Q/TOF-MS was combined with multivariate statistical analysis to identify the influence of paeonol on small molecule metabolites relevant to endometriosis. Finally, the key targets were screened using a combination of network pharmacology and molecular docking approaches. The results showed that the pathological indexes of rats were improved and returned to normal levels after treatment with paeonol, which was the basis for confirming the efficacy of paeonol. Metabolomics results identified 13 potential biomarkers, and paeonol callbacks 7 of them, involving six metabolic pathways. Finally, four key genes were found for paeonol therapy of endometriosis, and the results of molecular docking revealed a significant interaction between paeonol and the four key genes. This study was successful in establishing a rat model of endometriosis with cold coagulation and blood stagnation. GCH1, RPL8, PKLR, and MAOA were the key targets of paeonol in the treatment of endometriosis. It is also demonstrated that metabolomic techniques give the potential and environment for comprehensively understanding drug onset processes.

## 1. Introduction

Endometriosis is an estrogen dependent inflammatory disease that accounts for 10% to 15% of women in their reproductive years and up to 30–50% of women with infertility [1,2,3,4,5]. Endometriosis affects women significantly and its clinical manifestations are mainly infertility and chronic pelvic pain, which account for about 1/3 of infertile women and 2/3 of women with chronic pelvic pain [6,7]. Conventional therapies include danazol, progesterone, NSAIDs, etc. [8,9,10,11]. The drugs mentioned aim to lower estrogen levels, raise progesterone levels, or suppress inflammation [12]. Although drugs have shown good therapeutic effects, the following points should be considered when using them. First, high levels of estrogen induce potential side effects [13]. Second, the recurrence rate remains high. Third, there are huge medical costs [14,15]. However, Traditional Chinese Medicine (TCM) can be effective in avoiding these disadvantages to some extent [16,17].

In TCM theory, blood stasis is one of the factors leading to endometriosis. Symptoms of blood stasis include pain in a fixed location, a dark purple face or tongue, and bleeding [18]. Guizhi Fuling Decoction [19] can improve cold stagnation and blood stasis by promoting blood circulation and removing stasis to treat endometriosis. However, Guizhi Fuling Decoction, as a classical formula, has complex and unclear composition. Paeonol in Guizhi Fuling Decoction was found to improve disease by regulating inflammation, platelet aggregation, lipid metabolism, mitochondrial damage, and endoplasmic reticulum autophagy [20,21]. Peng Yi et al. investigated the mechanism by which paeonol alleviates dysmenorrhea in mice and found that paeonol reduces inflammatory cytokines and simultaneously inhibits uterine contractions [22,23]. Based on our earlier study, which found that Guizhi Fuling Decoction was successful in treating endometriosis with cold coagulation and blood stagnation, according to the content of paeonol in Guizhi Fuling Decoction, rats were given 4 times the quantity of paeonol, and it was discovered that 4 times the dose of paeonol had a significant effect [24].

In this study, we established a model of endometriosis with cold coagulation and blood stagnation. The efficacy of paeonol was confirmed by morphological examination (ectopic endometrial tissue), histological examination (HE staining), estradiol level detection, and progesterone level detection combined with metabolomics, network pharmacology, molecular docking for plasma biomarkers, and key targets screening.

## 2. Results

### 2.1. Morphological Observations and Histopathological Observations

The ectopic lesion volume showed a significant difference (Figure 1A, Table 1). We observed that the ectopic tissue in the model group showed a cystic vesicle with hyaline or translucent fluid inside. Fortunately, the volume of ectopic endometrial tissue became smaller or disappeared significantly after treatment with danazol and paeonol (compared to the model group).

The results of staining with H&E (Figure 1D) in the control group showed that the endometrial structure is intact. The glands were abundant and clearly structured. The glandular epithelial cells were tall columnar and arranged neatly and tightly, with abundant cytoplasm. The interstitial cells were in shuttle shape, neatly arranged, and evenly distributed. The blood vessels were abundant. In the model group, the proliferation of ectopic endometrial epithelial cells was short columnar, partially pseudostratified, and the cell morphology was incomplete. Compared with the control group, the number of interstitial cells and glands increased significantly, and blood vessels were abundant. When compared to the model group, the paeonol group showed varying degrees of improvement in the aforementioned pathology. The main manifestations were shrinkage of the cystic cavity, thinning of the cystic wall, short columnar or flat glandular epithelial cells, partial degeneration, necrosis or shedding, loose arrangement, irregular nuclei, and uncertain location. In addition, the number of glands decreased or even disappeared, showing atrophy. Inflammatory cells seemed to exudate. The number of interstitial cells decreased, or became smaller, sparsely arranged, or even non-existent. There were decreased blood vessels and fibrosis in some areas.

### 2.2. Changes in Plasma Estradiol and Progesterone Levels

According to the experimental results (Figure 1B,C and Table 1) estradiol levels were significantly increased after successful model preparation (*p* < 0.01). Estradiol levels were significantly lower in the paeonol and danazol groups compared to the model group (*p* < 0.01). Meanwhile, progesterone levels were significantly decreased (*p* < 0.01) after successful model preparation. The levels of progesterone were significantly higher in the paeonol and danazol groups compared with the model group (*p* < 0.01). This implies that paeonol affects estradiol and progesterone levels.

### 2.3. Evidence of Cold Clotting and Blood Stagnation in Rats

The rats in the model group displayed cold-causing disease symptoms such as poor mental condition, loose and lustrous dark fur, a tendency to pile up, lethargy and laziness, reduced food intake, and unformed feces. At the same time, the ears, paw nails, and tail were black, the blood vessels had swollen to a dark red color, and the veins under the tongue were slightly thick and dark purple, all of which were compatible with syndrome of cold clotting and blood stagnation in Chinese medicine. The paeonol, danazol, and control groups of rats had great mental health, lustrous fur, were responsive with normal diets and bowel motions, small and clear brilliant red blood vessels in the ears, a pink tongue, and normal subungual vein color.

### 2.4. Effects of Paeonol on Potential Plasma Biomarkers in Endometriosis Rats

Metabolomics analysis was performed on plasma samples obtained from control and model rats. To find endogenous metabolites that contributed significantly to intergroup clustering, the supervised pattern recognition method OPLS-DA was performed (ESI−: R^2^Y − 0.997, Q^2^ − 0.891; ESI+: R^2^Y − 0.884, Q^2^ − 0.810 (Figure 2A,D and Appendix A). Further analysis was performed to obtain S-plot load analysis plots (Figure 2B,E) and variable importance in the projection plots (Figure 2C,F), using VIP > 1 and *p* < 0.05 as the criteria for screening potential biomarkers. Finally, after a statistical analysis of the data, 13 plasma biomarkers associated with the rat model of endometriosis were identified, eight biomarkers in the positive ion mode and five biomarkers in the negative ion mode (Appendix A). To further investigate the trends of these biomarkers in control and model rats, we performed statistical analysis of the changes in the levels of the above screened and identified biomarkers in the control and model groups. The results are shown as an aggregated heat map (Figure 2G).

The results of PLS−DA analysis in the control, model, and paeonol groups showed that the paeonol group was close to the control group (ESI−: R^2^Y − 0.993, Q^2^ − 0.857; ESI+: R^2^Y − 0.994, Q^2^ − 0.896 (Figure 3A,B and Appendix A), suggesting the effectiveness of paeonol in treating endometriosis. Paeonol back-regulated seven plasma biomarkers associated with the rat model of endometriosis, namely L-phenylalanine, L-tryptophan, hippuric acid, LysoPC(18:4-(6Z,9Z,12Z,15Z)), LysoPE(0:0/20:4(8Z,11Z,14Z,17Z)), LysoPC(18:1(9Z)), and 5-HETE, specific information is shown in Figure 4 and Table 2. The seven identified plasma biomarkers were used for Met-PA analysis, suggesting that paeonol regulated six metabolic pathways, namely Phenylalanine metabolism, Aminoacyl-tRNA biosynthesis, Phenylalanine, tyrosine, and tryptophan biosynthesis, Arachidonic acid metabolism, Glycerophospholipid metabolism, Tryptophan metabolism. (Figure 3C and Table 3). Therefore, the metabolic pathways were identified as target pathways. The results of the specific data are shown in Table 3. The results suggest that these biomarkers and metabolic pathways are closely associated with paeonol for treating endometriosis.

### 2.5. The Therapeutic Targets of Paeonol on Endometriosis and Related Network Analysis

A total of 105 biomarker-related targets were obtained by MetPA analysis to build metabolic pathway-related gene networks. Additionally, 2750 disease targets were retrieved from the GeneCards database. The two were combined (Figure 5A) to get 32 therapeutic targets. The therapeutic targets were imported into the MetaboAnalyst 5.0 website and the DAVID database for KEGG and GO pathway enrichment analysis (Figure 5C,D and Table 4). Consequently, 13 KEGG pathways were identified, and five biomarkers were involved in these metabolic pathways. The five biomarkers, KEGG pathways, and related genes were also built as a correlation network diagram (Figure 5B). The five KEGG pathways that overlapped with metabolic pathways were phenylalanine metabolism; aminoacyl-tRNA biosynthesis; phenylalanine, tyrosine, and tryptophan biosynthesis; arachidonic acid metabolism and tryptophan metabolism. Therefore, the results of network pharmacology verified the metabolic pathway found based on metabolomics. Related genes Cytoscape 3.8.2 software was imported for topology analysis using the Network Analyzer tool and the top four genes of freedom were selected as key genes: GCH1, RPL8, PKLR, and MAOA.

The pdb codes of the target proteins GCH1, PKLR, RPL8, and MAOA were 6Z89, 7QDN, 4CCM, and 2Z5Y, respectively. Paeonol showed good affinity for selected core targets (Figure 6A–D and Table 5), with yellow dashed lines representing hydrogen bonds, amino acids bound to hydrogen bonds processed as green, and small molecule ligands processed as red. Molecular docking revealed that small molecules were all stably located within the docking pocket. These results suggest that these targets may be one of the mechanisms of paeonol in treating endometriosis.

## 3. Discussion

The study used autologous transplantation combined with a 28 day ice water bath to create a rat model of endometriosis with cold clotting and blood stagnation. This can not only meet the conditions of Western medicine disease model formation, but also have the characteristics of a “syndrome” in Chinese medicine. This is consistent with the theory of “dialectical treatment” in Chinese medicine.

In terms of plasma metabolites, plasma metabolomics was performed in rats with endometriosis models, and 13 differential metabolites were identified, 7 of which were paeonol callbacks. Based on this, further analysis combined with network pharmacology resulted in the identification of four key genes, namely GCH1, RPL8, PKLR, and MAOA.

Monoamine oxidase A (MAOA), as a flavoenzyme, can catabolize other neurotransmitters, including norepinephrine (NE) and serotonin (5HT) [25]. NE is a powerful *α* agonist that can elicit vasoconstriction via the agonistic impact of receptors. 5HT is a monoaminergic neurotransmitter that influences vasoconstriction [26]. Dysmenorrhea is caused by uterine vasoconstriction, and one of the clinical symptoms of endometriosis is dysmenorrhea. NE reduces cytokine expression in microglia, astrocytes, and brain endothelial cells, resulting in increased inflammation [27]. Serotonin release causes platelets to enhance inflammatory response. Endometriosis is a chronic inflammatory reaction. Serotonin is strongly associated with endometriosis, which is synthesized from the essential amino acid L-tryptophan. The content of L-tryptophan in the plasma of rats in the model group was reduced in the current study, and endometriosis had a tumor-like malignant proliferative capacity. It has been discovered that L-tryptophan metabolites can limit T-cell proliferation and induce death, resulting in an immunosuppressive effect. Excessive L-tryptophan catabolism is a major cause of immunological tolerance in endometriosis implantation; L-tryptophan levels in the plasma increased after paeonol treatment, and endometriosis improved.

GTP cycle hydrolase 1 (GCH1)-regulated miRNAs are involved in microglia activation and affect neuropathic pain [28]. GCH1, a member of the family of enzymes encoding GTP ring hydrolases, encodes a protein that is the first rate-limiting enzyme in tetrahydrobiopterin (BH4) biosynthesis [29]. BH4 is a cofactor of phenylalanine hydroxylase. Phenylalanine metabolism and its levels in the plasma are directly related to phenylalanine hydroxylase activity, while inflammation-induced reactive oxygen species (ROS) production consumes a large portion of BH4, leading to impaired phenylalanine degradation [30,31]. This is consistent with the elevated plasma phenylalanine levels in the model group of rats in this study. Under normal conditions, the expression of GCH1 is upregulated by the increase in phenylalanine [32]. GCH1 affects immunosuppression by influencing 5-HT levels in tryptophan metabolism and activating AHR and IDO1 and influences immunity by regulating T cell subsets [33,34]. Phenylalanine is involved in the metabolism of phenylalanine, and the biosynthesis of phenylalanine, tyrosine, and tryptophan. Phenylalanine metabolism is commonly thought to be associated with oxidative stress and inflammatory responses, and hippuric acid is an intermediate product of phenylalanine metabolism. Hippuric acid is converted to acylglycine along with the participating glycine and benzoic acid. The overproduction of acylglycine induces impaired mitochondrial fatty acid *β*-oxidation and is therefore involved in the inflammatory response. Moreover, during inflammation, the expression level of hippuric acid is increased to varying degrees [35]. The accumulation of hippuric acid in the body suggests that inflammation interferes with the metabolism of phenylalanine. Hippuric acid levels in the plasma decreased after paeonol treatment. Paeonol may exert anti-inflammatory and antioxidant effects by reducing hippuric acid and L-phenylalanine levels, thereby treating endometriosis.

Ribosomal protein L8 (RPL8) is a ribosomal protein, and RPL8 and its co-expressed genes are involved in the immune process and are markers of iron death, affecting development and immunity [36,37]. Retrograde menstruation is the transport of menstrual endometrial tissue through the fallopian tubes to the peritoneal cavity [38]. Iron overload may originate from pelvic erythrocyte lysis and exert significant cytotoxic effects on cells such as endometrial stromal cells [39]. Iron death of endometrial stromal cells triggers the production of angiogenic, inflammatory, and growth cytokines. In particular, angiogenic cytokines such as vascular endothelial growth factor A (VEGFA) and interleukin 8 (IL8) promote angiogenesis, inflammation, and growth cytokine production in human umbilical vein endothelial cells (HUVEC) in vitro. In addition, we found that inhibition of p38 mitogen-activated protein kinase/signal transducer and activator of transcription 6 (p38 MAPK/STAT6) signaling suppressed VEGFA and IL8 expression when iron death occurred in endometrial stromal cells, which may contribute to endometriotic lesion angiogenesis [40].

Pyruvate kinase (PKLR) catalyzes the transphosphorylation of phosphoenolpyruvate to pyruvate and ATP, a core metabolite of glycolytic metabolism. It mediates endometriosis apoptosis through inhibition of aerobic glycolytic metabolism to improve endometriosis [41]. Peritoneal fluid in women with endometriosis produces toxicity factors that increase with the severity of endometriosis [42]. Pyruvate reduces this toxic effect and provides a possible mechanism of infertility in endometriosis [43]. PKLR regulates glutathione levels and converts glutathione to reduced glutathione, which is involved in the regulation of oxidative stress damage in vivo [44]. Oxidative stress is involved in the development of endometriotic lesions via mitogen-activated protein (MAP) kinase or extracellular signal-regulated kinase (ERK) pathways, through the expression and action of intranuclear phosphorylated protein (c-Fos) and amino-terminal protein kinase (c-Jun) [45].

LysoPC(18:4(6Z,9Z,12Z,15Z)), LysoPC(18:1(9Z)), and LysoPE(0:0/20:4(8Z,11Z,14Z,17Z)) are glycerophospholipids, which are key ingredients of cellular lipid bilaterals and can be involved in metabolism and signaling, and studies have shown that glycerophospholipids are the main lipid metabolism pathway. Glycerophospholipid metabolism disorders reflect systemic changes caused by inflammatory response and oxidative stress; abnormalities in LysoPC(18:4(6Z,9Z,12Z,15Z)) LysoPC(18:1(9Z)), and LysoPE(0:0/20:4(8Z,11Z,14Z,17Z)) will accelerate developing endometriosis [46]. Glycerophospholipids induce cyclotoxinase-2 (COX-2), which catalyzes the conversion of arachidonic acid, which plays an important role in female reproduction. Metabolites of arachidonic acid are involved in various reproductive activities such as luteolysis, endometrial gene expression, and development [47]. Arachidonic acid metabolism plays a crucial role in the inflammatory process [48]. Alammari Ahmad H et al. found that angiotensin II can be inhibited by suppressing the associated HETE [49]. Angiotensin II, a major effector of the renin–angiotensin system, is present in endometrial tissue and involved in cyclic changes in the endometrium [50]. Angiotensin II has also been shown to be an important factor in promoting physiological and pathological angiogenesis [51]. The key to the implantation, invasion, and growth of endometriotic lesions is maintaining blood supply through neointima formation, which promotes the development and recurrence of endometriosis. Treating endometriosis by inhibiting the neovascularization of ectopic lesions has become a hot research topic in this field. In our experiments, 5-HETE levels were reduced in the model group compared to the control group. Fortunately, paeonol treatment elevated the level of 5-HETE, which proved the inference correct and justified the inference.

Paeonol has a wide range of biological activities, such as anti-inflammatory, antioxidant, analgesic, immune improvement, and antidepressant [52,53]. In pharmacokinetic studies, paeonol is rapidly absorbed into the blood and widely distributed in various tissues. At the same time, paeonol is involved in hepatic and intestinal circulation, so the blood concentration is stable and the duration of drug action is prolonged [54]. The above evidence shows that paeonol treats endometriosis by regulating immunity, inflammation, and antioxidants.

However, the key targets, core biomarkers, and metabolic pathways obtained in this study were not experimentally validated, and further experimental validation of the therapeutic targets of paeonol for endometriosis is needed in future studies.

## 4. Materials and Methods

### 4.1. Chemicals and Reagents

Paeonol (99.9%) was supplied by Chengdu Pusi Biotechnology Co., Ltd. (Chengdu, China. Batch number PS160806-01). Danazol Capsules was supplied by Jiang su lian huan Pharmaceutical Co., Ltd. (Yangzhou, China. batch number H20023116), and ELISA Kit were obtained from R&D Systems (Minneapolis, MN, USA. Batch number E20180101A). Estradiol benzoate injection was supplied by Animal Pharmaceutical Hangzhou. (Hangzhou, China. batch number 110202511). Distilled water was obtained from Watsons Water (Guangzhou, China). Acetonitrile and methanol were supplied by Fisher (Loughborough, UK), while formic acid and leucine enkephalin were purchased from Sigma–Aldrich (St Louis, MO, USA).

### 4.2. Animals and Groups

Female Wistar rats, weighing 180–220 g, were obtained from the GLP Center of the Heilongjiang University of Chinese Medicine (Permit No. SCXK, 2014-004). The room temperature and relative humidity were controlled at the range of 22–26 °C and 35–45% respectively, with a 12 h light or dark cycle. Before the experiment, rats were acclimatized and fed for 7 days, and the rats had free access to water and food. Then, all rats were randomly divided into four groups: the sham operation control group (control), the endometriosis model group (model), the paeonol group (paeonol), and the danazol group (danazol). All experiments were performed following the Declaration of Helsinki and were approved by the Animal Health and Ethics Committee of Heilongjiang University of Traditional Chinese Medicine (2021012709).

### 4.3. Preparation of Animal Models and Paeonol Solution

An autologous graft combined with an ice water bath was used to prepare an endometriosis model, and the animal experimental procedure was performed during estrus. Rats were anesthetized with sodium pentobarbital (0.2 mL/100 g) and fixed under aseptic conditions, and the uterine horns were cut into two pieces of approximately 4 mm × 4 mm. Then, the tissue was sutured into the uterus and abdominal wall of the same rat. Next, gentamicin (0.5 mL per rat) was injected into the rats daily for three days for anti-inflammatory and anti-infection. Later, estradiol (0.1 mL per rat) was injected subcutaneously every other day three times a day to help recover the rats’ uteruses. Rats in the control group had only one side of the uterine horn removed without transplantation. After two weeks of recovery, the rats in the model, paeonol, and danazol group were immersed in ice water (0–1 °C) for 8–10 min, lasting for two weeks. The paeonol group was given 344.04 mg/kg paeonol by gavage (an oral solution made by dissolving paeonol standard powder in 1% CMC-Na solution). The danazol group was given 72 mg/kg danazol by gavage. The other two groups of rats were given 1% CMC-Na solution in the same manner. The treatment course lasted for four weeks.

### 4.4. Calculation of the Volume of the Ectopic Tissue

On the last day, the rats were anesthetized with sodium pentobarbital (0.2 mL/100 g) after 60 min of treatment. The ectopic tissue was removed with a surgical blade, multiplied the length, breadth, and height of the ectopic tissue measured with a vernier caliper by a factor of 0.52, then the result was multiplied by a factor of 0.52 to achieve the transplant volume.

### 4.5. Hematoxylin-Eosin Staining (H&E Staining)

The excised ectopic endometrium was cut into 1 × 1 mm pieces and promptly fixed in 10% formaldehyde before being embedded in paraffin and sectioned. After deparaffinization, the slices were stained with hematoxylin and eosin and dried for histological investigation. Using a microscope, the changes in the morphological features of each group were observed.

### 4.6. Enzyme-Linked Immunosorbent Assay (ELISA)

The blood was collected from the abdominal aorta 60 min after the final day of treatment, centrifuged (3000 rpm for 15 min), and the supernatant was frozen at −80 °C for detection. ELISA was used to measure the levels of estradiol (E2) and progesterone (P) in rat plasma.

### 4.7. Preparation and Analysis of Plasma Samples

Before UPLC-MS analysis, 100 μL of plasma sample thawed on ice was added to 300 μL of methanol and vortex mixed to remove proteins. After standing at 4 °C for 15 min, it was centrifuged at 13,000 rpm for 10 min. The supernatant was taken and dried with nitrogen. A total of 200 μL of methanol solution was added for precipitation. It was centrifuged again at 13,000 rpm for 10 min at 4 °C. Finally, it was transferred to a liquid phase vial for UPLC-MS analysis.

### 4.8. Chromatography and Mass Spectrometry Conditions

#### 4.8.1. Chromatographic Conditions

Chromatographic separation was performed on a Water Acquity UPLC (Waters Corp., Milford, MA, USA). We used a Waters Acquity UPLC Ethylene Bridged Hybrid (BEH) C_18_ column (2.1 nm × 50 mm, 1.7 µm) at 40 °C. Mobile phases were acetonitrile with 0.1% formic acid (A) and water with 0.1% formic acid (B). The flow rate of the mobile phase was 0.40 mL/min. The injection volume in positive ion mode was 3 µL and in negative ion mode was 2 µL. The gradient program was as follows: 98–60%B (0–4 min), 60–30%B (4–10 min), 30–0%B (10–13 min).

#### 4.8.2. Mass Spectrometry Conditions

MS data were acquired by using a Waters LCT Premier XE TOF-MS (Waters, Mircomas, MA, USA). Positive and negative ionization modes were acquired via DuoSpray electrospray ionization (ESI). Optimal conditions for high-resolution MS analysis were as follows: Negative ion mode: capillary voltage 1.5 kV, cone voltage 50 V, ion source temperature 110 °C, dissolvent gas temperature 350 °C, dissolvent gas flow rate 750 L/h. Positive ion mode: capillary voltage 1.3 kV, cone voltage 45V, ion source temperature 110 °C, dissolvent gas temperature 350 °C, dissolvent gas flow rate 750 L/h. For accurate mass calibration application, Waters Lockspary calibration system was used for online real-time calibration with leucine-enkephalin solution ([M+H]^+^ = 556.2771, [M−H]^−^ = 554.2620), with a calibration solution injection rate of 100 µL/min and calibration frequency of 5 s. The scan mode was full scan, mass data acquisition range of m/z 100–1500. The scan time was 0.2 s, and data acquisition was performed in centroid mode.

### 4.9. Metabolomics Analysis

The ESI-MS metabolic profiles (.raw data) obtained by the UPLC-MS system were imported into Progenesis QI software (V 2.1, Waters Corporation, Milford, MA, USA) for comparison, peak extraction, and normalization. Then, the data matrix (.usp data) was obtained by multivariate statistical analysis using SIMAC software (V 14.1). OPLS-DA analysis was performed on the metabolic data matrices of the control and model groups. Potential biomarkers were screened with VIP > 1 and *p* < 0.05. Putative metabolites were derived by searching accurate molecular mass data from redundant m/z peaks through the HMDB (https://www.hmdb.ca/) and KEGG (https://www.genome.jp/kegg/) databases. The identified metabolites were then validated using an in-house fragment library. The changes of these potential biomarkers in control, model, and paeonol groups were statistically analyzed. The biomarkers of paeonol back regulation were found and imported into the website MetaboAnalyst 5.0 (http://www.metabioanalyst.ca) for metabolic pathway analysis.

### 4.10. Construction and Analysis of Hey Biomarker Biological Networks

Biomarkers regulated by paeonol were imported into MetaboAnalyst 5.0 (https://www.metaboanalyst.ca/, accessed on 25 September 2022) to build a metabolite-gene network and identify metabolite-related genes. To find disease genes, the keyword “endometriosis” was entered into the GeneCards (https://www.genecards.org/, accessed on 25 September 2022) database, and Venn diagrams were used to identify the intersection of metabolite-related genes and disease genes. The common genes obtained from the Venn diagram were imported into MetaboAnalyst 5.0 for KEGG analysis, into the DAVID database (https://david.ncifcrf.gov/) for GO analysis, and into STRING (https://string-db.org/2022, accessed on 25 September 2012) online database for analysis, and built a protein–protein interaction (PPI) network. The network was imported into Cytoscape 3.8.2 software (https://cytoscape.org/, accessed on 26 September 2022). Topology analysis was performed using the Network Analyzer tool to select the top four genes of freedom as potential targets of paeonol for treating endometriosis. The core targets screened by network topology analysis were used as receptors and paeonol as ligands for molecular docking. The 3D structure of paeonol was obtained through the Pubchem database (https://pubchem.ncbi.nlm.nih.gov/, accessed on 28 September 2022), saved as .sdf format, and the corresponding .pdbqt format was saved after adding polar hydrogen atoms to paeonol small molecules using AutoDockTool 1.5.6 software. Then, core target receptors were obtained from the PDB database (https://www.rcsb.org/, accessed on 28 September 2022), the crystal structure of the protein receptor was obtained to preserve .pdb format, water molecules and pre-existing ligands were removed using Pymol 2.3.0 software, and docking active sites were obtained using plugins and saved as .pdb format. The corresponding .pdbqt format was saved by AutoDockTool 1.5.6 software after adding polar hydrogen to the receptor. Importing AutoDock Vina (http://vina.scripps.edu/), molecular docking was performed, and finally, the docked complexes were visualized using Pymol 2.4 software.

### 4.11. Statistical Analysis

Data were tested by *t*-test and expressed as mean ± SD. differences in means were calculated to be statistically significant, *p* < 0.05 was considered statistically significant, and *p* < 0.01 implied a highly significant difference.

## 5. Conclusions

In this study, a rat model of endometriosis with cold clotting and blood stagnation was successfully established. The therapeutic effect of paeonol on endometriosis was demonstrated by ectopic tissue morphology, histopathological indices, enzyme-linked immunosorbent assay, and plasma metabolomics. The combination of metabolomics, network pharmacology, and molecular docking techniques elucidated the involvement of paeonol in the treatment of endometriosis. The plasma metabolic pathways involved in paeonol’s treatment of endometriosis were phenylalanine metabolism; aminoacyl-tRNA biosynthesis; phenylalanine, tyrosine and tryptophan biosynthesis; arachidonic acid metabolism; glycerophospholipid metabolism; and tryptophan metabolism. Core plasma biomarkers were L-phenylalanine, L-tryptophan, hippuric acid, LysoPC(18:4(6Z,9Z,12Z,15Z)), LysoPE(0:0/20:4(8Z,11Z,14Z,17Z)), LysoPC(18:1(9Z)), and 5-HETE; core targets were GCH1, RPL8, PKLR, and MAOA. This provides a scientific basis for paeonol to treat the disease. The network pharmacology results confirmed that metabolomics is an important tool for exploring the therapeutic mechanisms of drugs in diseases. However, the above key targets need further experimental validation.

## Figures and Tables

**Figure 1 molecules-28-00653-f001:**
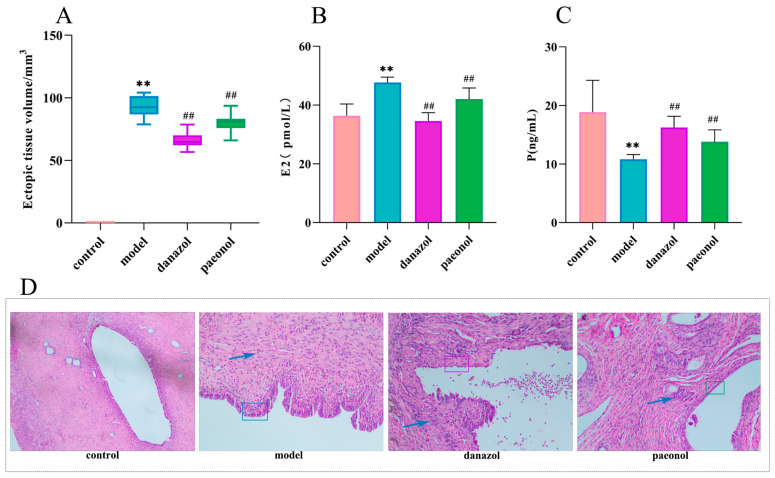
(**A**). Changes in ectopic lesion volume in the control, model, danazol, and paeonol groups. ** *p* < 0.01 compared to the control group; ^##^
*p* < 0.01 compared to the model group (*n* = 10). (**B**). Changes in plasma estradiol levels in the control group, model group, danazol group, and paeonol group. ** *p* < 0.01 compared to the control group; ^##^
*p* < 0.01 compared to the model group (*n* = 10). (**C**). Changes in progesterone levels in the control group, model group, danazol group, and paeonol group. ** *p* < 0.01 compared to the control group; ^##^
*p* < 0.01 compared to the model group (*n* = 10). (**D**). HE staining (×200) showed ectopic lesions in the control group, model group, danazol group, and paeonol group.

**Figure 2 molecules-28-00653-f002:**
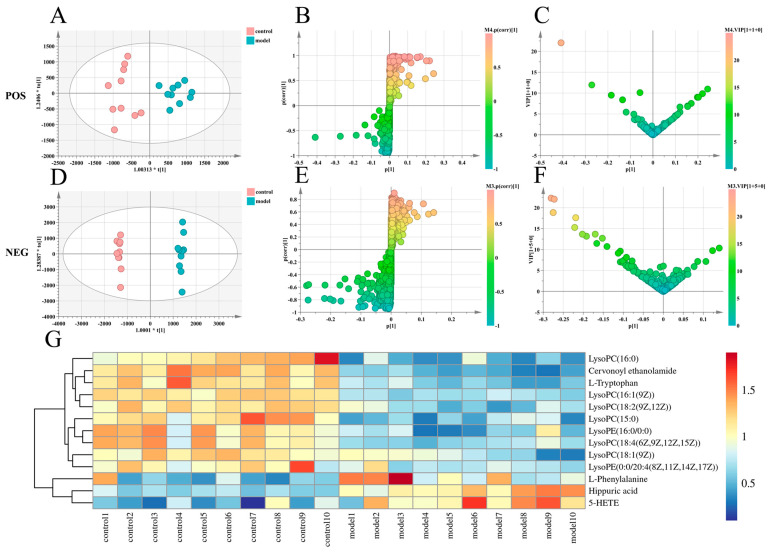
Multivariate statistical analysis. (**A**). OPLS−DA score plots for the control group and the model group in positive ion mode. (**B**). Plasma biomarkers in the S-plot between the control group and the model group in positive ion mode. (**C**). Plasma biomarkers in the VIP between the control group and the model group in positive ion mode. (**D**). OPLS−DA score plots for the control group and the model group in negative ion mode. (**E**). Plasma biomarkers in the S−plot between the control group and the model group in negative ion mode. (**F**). Plasma biomarkers in the VIP between the control group and the model group in negative ion mode. (**G**). Heatmap of changes in levels of 13 potential biomarkers in control and model groups. (OPLS−DA, orthogonal partial least squares discriminant analysis; VIP, variable importance in projection).

**Figure 3 molecules-28-00653-f003:**
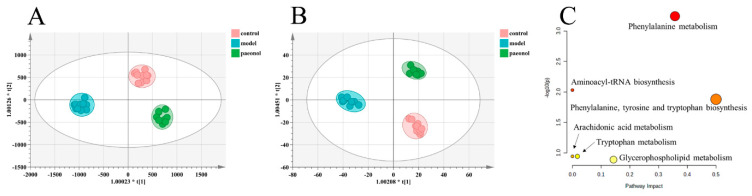
(**A**). PLS−DA score plots for the control, model, and paeonol groups in the positive ion mode. (**B**). PLS−DA score plots for the control, model, and paeonol groups in the negative ion mode. (**C**). Metabolism pathway analysis of seven biomarkers of paeonol back regulation with MetPA. The size and color of each circle are based on pathway impact value and *p*-value, respectively. (PLS−DA, Partial least squares Discriminant Analysis).

**Figure 4 molecules-28-00653-f004:**
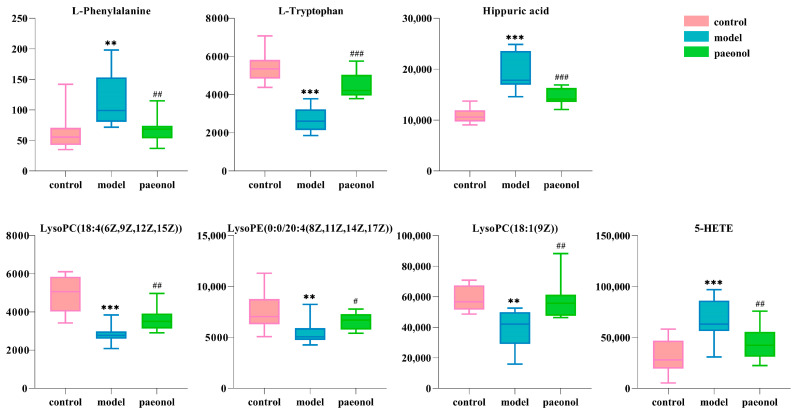
Box plots of the changes in the seven biomarkers of paeonol back regulation in the control, model, and paeonol groups. ** *p* < 0.01, *** *p* < 0.001 compared to the control group; ^#^
*p*<0.05, ^##^
*p* < 0.01, ^###^
*p* < 0.001 compared to the model group (*n* = 10).

**Figure 5 molecules-28-00653-f005:**
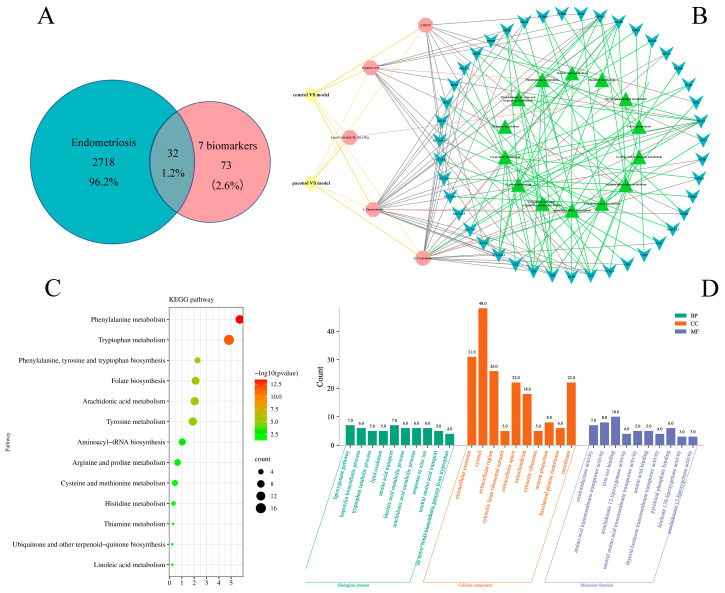
(**A**). Venn diagram of 32 intersection targets of 105 biomarker-related targets and 2750 disease targets. (**B**). Biological network diagram of biomarker-pathway gene. The yellow solid line connects biomarkers with elevated levels in the plasma compared to the model group, the yellow dashed line connects biomarkers with reduced levels in the plasma compared to the model group, the pink line connects biomarkers and metabolic pathways, the gray line connects biomarkers and genes, and the green line connects genes and metabolic pathways. (**C**). The 13 KEGG pathways associated with seven biomarkers. (**D**). Biomarker-related GO enrichment pathways.

**Figure 6 molecules-28-00653-f006:**
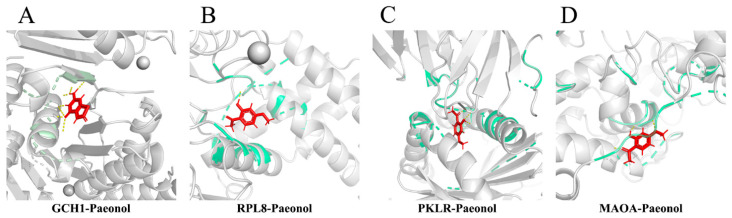
The molecular docking pattern diagram (**A**). Binding mode of GCH1 with Paeonol. (**B**). Binding mode of RPL8 with Paeonol. (**C**). Binding mode of PKLR with Paeonol. (**D**). Binding mode of MAOA with Paeonol.

**Table 1 molecules-28-00653-t001:** Changes in ectopic endometrial tissue volume and changes in estradiol and progesterone in plasma ($\bar{X}$ ± SD).

Group	Ectopic Tissue Volume/mm^3^	*p* (ng/mL)	E2 (p·mol/L)
Control group	0.00 ± 0.00	18.86 ± 5.43	36.36 ± 4.02
Model group	92.87 ± 8.54 **	10.80 ± 0.83 **	47.68 ± 1.84 **
Danazol group	66.08 ± 6.36 ^##^	16.25 ± 1.91 ^##^	34.57 ± 2.85 ^##^
Paeonol group	80.07 ± 7.12 ^##^	13.80 ± 2.04 ^##^	42.03 ± 3.78 ^##^

Note: Data are expressed as means ± SD. Note: Compared with the control group, ** *p* < 0.01; compared with the model group, ^##^
*p* < 0.01 (*n* = 10).

**Table 2 molecules-28-00653-t002:** Information on seven plasma potential biomarkers for paeonol callback.

No.	Rt min	Determined Mass	Calc Mass	[M−H]^−^/[M+H]^+^	ppm	Proposed Composition	Postulated Identity	HMDB Number	Trend in Model	Paeonol
1	1.36	166.0869	166.0868	[M+H]^+^	0.6	C_9_H_11_NO_2_	L-Phenylalanine	HMDB0000159	↑ **	↓ ^##^
2	1.73	203.0827	203.0821	[M−H]^−^	3.0	C_11_H_12_N_2_O_2_	L-Tryptophan	HMDB0000929	↓ ***	↑ ^###^
3	2.20	178.0504	178.0504	[M−H]^−^	0.0	C_9_H_9_NO_3_	Hippuric acid	HMDB0000714	↑ ***	↓ ^###^
4	4.79	514.2930	514.2934	[M−H]^−^	−1.0	C_26_H_46_NO_7_P	LysoPC(18:4(6Z,9Z,12Z,15Z))	HMDB0010389	↓ ***	↑ ^##^
5	7.71	500.2790	500.2777	[M−H]^−^	2.4	C_25_H_44_NO_7_P	LysoPE(0:0/20:4(8Z,11Z,14Z,17Z))	HMDB0011488	↓ **	↑ ^#^
6	8.64	522.3563	522.3560	[M+H]^+^	0.6	C_26_H_52_NO_7_P	LysoPC(18:1(9Z))	HMDB0002815	↓ **	↑ ^##^
7	8.99	319.2278	319.2273	[M−H]^−^	1.6	C_20_H_32_O_3_	5-HETE	HMDB0011134	↑ ***	↓ ^##^

Note: ↑ and ↓ indicate that the level of the biomarker is increased or decreased, respectively. ** *p* < 0.01, *** *p* < 0.001 compared to the control group; ^#^
*p* < 0.05, ^##^
*p* < 0.01, ^###^
*p* < 0.001 compared to the model group.

**Table 3 molecules-28-00653-t003:** Information on pathways involved in paeonol callback of seven biomarkers.

Pathway	Total	Hits	Raw *p*	FDR	Impact
Phenylalanine metabolism	12	2	0.000572	0.04808	0.35714
Aminoacyl-tRNA biosynthesis	48	2	0.009322	0.36963	0.00000
Phenylalanine, tyrosine, and tryptophan biosynthesis	4	1	0.013201	0.36963	0.50000
Arachidonic acid metabolism	36	1	0.113870	1.00000	0.00000
Glycerophospholipid metabolism	36	1	0.113870	1.00000	0.01736
Tryptophan metabolism	41	1	0.128830	1.00000	0.14305

**Table 4 molecules-28-00653-t004:** Information on KEGG pathways associated with seven biomarkers.

Pathway	PVal.Z	*p* Value	Hits
Phenylalanine metabolism	5.700	4.08 × 10^−14^	11
Tryptophan metabolism	4.820	3.50 × 10^−12^	16
Phenylalanine, tyrosine, and tryptophan biosynthesis	2.270	1.52 × 10^−6^	5
Folate biosynthesis	2.110	3.31 × 10^−6^	9
Arachidonic acid metabolism	2.040	4.73 × 10^−6^	10
Tyrosine metabolism	1.890	1.01 × 10^−5^	10
Aminoacyl-tRNA biosynthesis	1.030	0.000794	7
Arginine and proline metabolism	0.649	0.005540	6
Cysteine and methionine metabolism	0.437	0.016300	5
Histidine metabolism	0.321	0.029300	3
Thiamine metabolism	0.290	0.034400	2
Ubiquinone and other terpenoid-quinone biosynthesis	0.218	0.049400	2
Linoleic acid metabolism	0.218	0.049400	2

**Table 5 molecules-28-00653-t005:** The key target of paeonol for the treatment of endometriosis and paeonol molecular docking score.

**Paeonol**	**Target (pdb Code)**	**Combination of Free Energy**
	GCH1 (6Z89)	−5.300 kcal/mol
	RPL8 (4CCM)	−6.101 kcal/mol
	PKLR (7QDN)	−7.397 kcal/mol
	MAOA (2Z5Y)	−8.013 kcal/mol

## Data Availability

Not applicable.

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
