# Peer review of "UPLC-Q-TOF/MS Based Plasma Metabolomics for Identification of Paeonol’s Metabolic Target in Endometriosis"

_molecules, 2023, doi:10.3390/molecules28020653_

Round 1

Reviewer 1 Report

1. Introduction and throughout. TCM should be defined.

2. Figure 1 and throughout. H&E naming is inconsistent.

3. Section 2.4.  OPLS-DA is used for separation of groups, but no mention of PCA results either in text or supplemental material to determine of use of OPLS-DA is appropriate to use or if the results are over-fit.  

4. Figure 2G.  Heatmap shows hierarchal clustering between groups, but no differences observed in the heatmap.  Related p-values for each potential biomarker should be included. 

5. Figure 3.  See comments regarding OPLS-DA above.

6.  Text in figures is very difficult to read.  Re-labeling is strongly suggested.

7. Stereochemistry of amino acids was not conducted.  If it was, it was not included.  Differentiation between D- and L- metabolites should be explained in methods if performed.  Otherwise, that annotation should be removed from text.

8. Methods section is missing the instrumentation used for LC-MS.  This needs to be included along with how the data were processed from their raw form.  Were the data normalized and aligned?  Was the data filtered prior to statistical analysis?

9. Section 4.9.  "Compare 417 the changes in the levels of these potential markers in the control, model, and paeonol 418 groups".  This sentence is fragmented and does not make sense.  Please rewrite.

10. Supplemental material.  Please correct the missing information.  This section is incomplete.

Author Response

Dear reviewers and editors.

First of all, thank you very much for taking the time out of your busy schedule to read and revise my manuscript. Thank you for your valuable suggestions. You have made comprehensive corrections to the structure, content, research methods and results of my paper. It has played a very important role in improving the quality of my paper.

I have carefully studied the reviewers' comments and carefully revised the manuscript according to the suggestions as follows.

Point 1. Introduction and throughout. TCM should be defined.

Response 1: We are grateful for the suggestion. We have modified this expression throughout the manuscript according to the comment, such as"Traditional Chinese Medicine (TCM)" on page 2 lines 42.

Point 2. Figure 1 and throughout. H&E naming is inconsistent.

Response 2: Thank you for the constructive comments on my manuscript. The results of H&E were relabeled in the manuscript.

Point 3. Section 2.4. OPLS-DA is used for separation of groups, but no mention of PCA results either in text or supplemental material to determine of use of OPLS-DA is appropriate to use or if the results are over-fit.

Response 3: We appreciate the reviewer’s positive evaluation of our work. According to the reviewer’s comment, we have provided more details to describe the results are not over-fit on page 4 lines 115. “(ESI-: R2Y-0.997, Q2-0.891; ESI +: R2Y-0.884, Q2-0.810)”. Statistical validation of the OPLS-DA material using permutation analysis (Figure S1. (A) in positive mode supplemental analysis; (B) in negative mutation mode) was provided in supplemental material.

Point 4. Figure 2G. Heatmap shows hierarchal clustering between groups, but no differences observed in the heatmap. Related p-values for each potential biomarker should be included.

Response 4: Thank you for underlining this deficiency. Figure 2G was revised and modified according to the information suggested by the reviewer(On page 4. Figure 2G).

Point 5. Figure 3. See comments regarding OPLS-DA above.

Response 5: Thank you for the constructive comments on my manuscript. According to the reviewer’s comment, we have provided more details to describe the results are not over-fit on page 4 lines 135-136 “(ESI-: R2Y-0.993, Q2-0.857; ESI+: R2Y-0.994, Q2-0.896)”. Statistical validation of the PLS-DA material using permutation analysis (Figure S2. (A) in positive mode supplemental analysis; (B) in negative mutation mode) was provided in supplemental material.

Point 6. Text in figures is very difficult to read. Re-labeling is strongly suggested.

Response 6: We are grateful for the suggestion. We have modified this expression of text in figures throughout the manuscript according to the comment.

Point 7. Stereochemistry of amino acids was not conducted. If it was, it was not included. Differentiation between D- and L- metabolites should be explained in methods if performed. Otherwise, that annotation should be removed from text.

Response 7: We are extremely grateful to reviewer for pointing out this problem. Although stereochemistry of amino acids was not conducted, the endogenous amino acid should be performed in the L conformation. So we finally identified this amino acid as L- metabolites.

Point 8. Methods section is missing the instrumentation used for LC-MS. This needs to be included along with how the data were processed from their raw form. Were the data normalized and aligned? Was the data filtered prior to statistical analysis?

Response 8: We deeply appreciate the reviewer’s suggestion. Instrument information was added in the manuscript on page 11 lines 347 and354.

Chromatographic separation was performed on a Water Acquity UPLC (Waters Corp., Milford, MA, USA). We used an Waters Acquity UPLC Ethylene Bridged Hybrid (BEH) C18 column (2.1 nm×50 mm, 1.7 µm) at 40°C. Mobile phases were acetonitrile with 0.1% formic acid (A) and water with 0.1% formic acid (B). The flow rate of the mobile phase was 0.40 ml/min. The injection volume in positive ion mode was 3µl and in negative ion mode was 2µl. The gradient program was as follows: 98-60%B(0-4 min), 60-30%B(4-10 min), 30-0%B(10-13 min).

MS data were acquired by using an Waters LCT Premier XE TOF-MS (Waters, Mircomas, MA, USA). Positive and negative ionization modes were acquired via DuoSpray electrospray ionization (ESI). Optimal conditions for high resolution MS analysis were as follows: Negative ion mode: capillary voltage 1.5 kV, cone voltage 50 V, ion source temperature 110°C, dissolvent gas temperature 350 °C, dissolvent gas flow rate 750 L/h. Positive ion mode: capillary voltage 1.3 kV, cone voltage 45V, ion source temperature 110 °C, dissolvent gas temperature 350 °C, dissolvent gas flow rate 750 L/h. Accurate mass calibration applica-tion Waters Lockspary calibration system was used for online real-time calibration with leucine-enkephalin solution ([M+H]+=556.2771, [M-H]-=554.2620), calibration solution injection rate of 100 uL/min, calibration frequency of 5 s. The scan mode was full scan, mass data acquisition range of m/z 100-1500, The scan time was 0.2 s, and data acquisition was performed in centroid mode.

The ESI-MS metabolic profiles (.raw data) obtained by the UPLC-MS system were imported into Progenesis QI software (V2.1, Waters Corporation, MA) for comparison, peak extraction, and normalization.

Point 9. Section 4.9. "Compare 417 the changes in the levels of these potential markers in the control, model, and paeonol 418 groups". This sentence is fragmented and does not make sense. Please rewrite.

Response 9: Thank you for the constructive comments on my manuscript. We agree with the comment and re-wrote the sentence in the revised manuscript on page 11 lines 374 as the following: “The changes of these potential biomarkers in control, model and paeonol groups were statistically analyzed”.

Point 10. Supplemental material. Please correct the missing information. This section is incomplete.

Response 10: We are grateful for the suggestion. In order to improve the quality of our manuscript. This section is completed.

Finally, I would like to thank you again for your guidance and for reviewing and revising my manuscript again. I hope that under your guidance I can complete this excellent paper and sincerely hope that my paper will be published in your journal.

Reviewer 2 Report

Dear Authors

The manuscript with the title: UPLC-Q-TOF/MS-based Plasma Metabolomics for Identification of Metabolic Targets of Paeonol in Endometriosis has been well written and the results are interesting. However, some suggestions as below:

1. On page 2 lines 59-63, I do not see any reference to previous work. Please provide references

2. Compared to the model group, paenol and danazol are better at reducing estrogen levels and increasing progesterone levels. But what if paenol is compared to danazol? I think the activity of paenol is lower than danazol

3. Progesterone is also estrogen (page 2 lines 68-69)? I think the words should be separated 

4.  Regarding the docking procedure, I didn't see the pdb code and how the validation was done.

5. It needs to be further explained how the pdb is selected based on the selected gene. Is there a homology modeling procedure for protein targets? 

Author Response

Dear reviewers and editors.

First of all, thank you very much for taking the time out of your busy schedule to read and revise my manuscript. Thank you for your valuable suggestions. You have made comprehensive corrections to the structure, content, research methods and results of my paper. It has played a very important role in improving the quality of my paper.

I have carefully studied the reviewers' comments and carefully revised the manuscript according to the suggestions as follows.

Point 1. On page 2 lines 59-63, I do not see any reference to previous work. Please provide references.

Response 1: We are extremely grateful to reviewer for pointing out this problem. We have referenced [24] in the manuscript on page 2 lines 56.

[24] Xiuhong, W.; Fang, H.; Chengyu, P.; Dongxia, Y.; Fashan, W.; Luwen, H.; Zhang, N. High Performance Liquid Chromatography Fingerprint of Guizhi Fuling Pill. Fen Hsi K'o Hsueh Hsueh Pao 2014, 30, 100-102, doi:10.13526/j.issn.1006-6144.2014.01.022.

Point 2. Compared to the model group, paenol and danazol are better at reducing estrogen levels and increasing progesterone levels. But what if paenol is compared to danazol? I think the activity of paenol is lower than danazol.

Response 2: We thank the reviewers for their very precious comments. In fact, we found that paeonol was the pharmacodynamic material basis of Guizhi Fuling decoction in the treatment of endometriosis, and then pharmacodynamic and metabolomic studies of paeonol were performed. However, there was no particular focus in this study on the comparison of the efficacy of danazol and paeonol in the treatment of endometriosis, and we were very interested in conducting further studies on this, especially whether paeonol could compensate for some aspects of danazol deficiency.

Point 3. Progesterone is also estrogen (page 2 lines 68-69)? I think the words should be separated.

Response 3: Thank you for your suggestion. As suggested by reviewer, we have modified the suggested content to the manuscript on page2 lines 59: “estrogen level tests (estradiol and progesterone)” modified to “estradiol level detection, progesterone level detection”.

Point 4. Regarding the docking procedure, I didn't see the pdb code and how the validation was done.

Response 4: Thank you for the constructive comments on my manuscript. We are grateful for the suggestion. As suggested by the reviewer, we have added more details of the pdb codes of GCH1, RPL8, PKLR, and MAOA were 6Z89, 4CCM, 7QDN, and 2Z5Y, respectively. And pbd codes are supplemented in manuscript on page 8 lines 184-185. And we have added the validation process in the manuscript on page 12 lines 393-403. The validation process is as follows:

The 3D structure of Paeonol was obtained through Pubchem database (https://pubchem.ncbi.nlm.nih.gov/ accessed on September 28, 2022), saved as .sdf format, and the corresponding .pdbqt format was saved after adding polar hydrogen atoms to pae-onol small molecules using AutoDockTool 1.5.6 software. Then, core target receptors were obtained from the PDB database (https://www.rcsb.org/ accessed on September 28, 2022), the crystal structure of the protein receptor was obtained to preserve .pdb format, water molecules and pre-existing ligands were removed using Pymol 2.3.0 software, and docking active sites were obtained using Plugin plugins and saved as .pdb format. The correspond-ing .pdbqt format was saved by AutoDockTool 1.5.6 software after adding polar hy-drogenogen to the receptor. Importing AutoDock Vina (http://vina.scripps.edu/), molecular docking was performed, and finally, the docked complexes were visualized by Pymol soft-ware.

Point 5. It needs to be further explained how the pdb is selected based on the selected gene. Is there a homology modeling procedure for protein targets?

Response 5: Thank you again for your valuable comments. We chose the protein structure with the smallest refinement resolution and species of Homo sapiens as tagert in PDB database. There are suitable protein models in the PDB database, so homology modeling was not procedured.

Finally, I would like to thank you again for your guidance and for reviewing and revising my manuscript again. I hope that under your guidance I can complete this excellent paper and sincerely hope that my paper will be published in your journal.